# MODEL-BASED INFERENCE OF SYNAPTIC PLASTICITY RULES

## ABSTRACT

Understanding learning through synaptic plasticity rules in the brain is a grand challenge for neuroscience. Here we introduce a novel computational framework for inferring plasticity rules from experimental data on neural activity trajectories and behavioral learning dynamics. Our methodology parameterizes the plasticity function to provide theoretical interpretability and facilitate gradient-based optimization. For instance, we use Taylor series expansions or multilayer perceptrons to approximate plasticity rules, and we adjust their parameters via gradient descent over entire trajectories to closely match observed neural activity and behavioral data. Notably, our approach can learn intricate rules that induce long nonlinear time-dependencies, such as those incorporating postsynaptic activity and current synaptic weights. We validate our method through simulations, accurately recovering established rules, like Oja's, as well as more complex hypothetical rules incorporating reward-modulated terms. We assess the resilience of our technique to noise and, as a tangible application, apply it to behavioral data from *Drosophila* during a probabilistic reward-learning experiment. Remarkably, we identify an active forgetting component of reward learning in flies that enhances the predictive accuracy of previous models. Overall, our modeling framework provides an exciting new avenue to elucidate the computational principles governing synaptic plasticity and learning in the brain.

## 1 INTRODUCTION

While the backpropagation algorithm has been a cornerstone in the training of artificial neural networks (Rumelhart et al., 1986), its biological implausibility limits its relevance for biology (Lillicrap et al., 2020). Specifically, backpropagation involves non-local weight updates and symmetric feedback connections, mechanisms that are not observed in biological neural systems. This leads to a compelling question: how do biological brains, devoid of backpropagation, accomplish learning?

Synaptic plasticity, the ability of synapses to change their strength, is a key neural mechanism underlying learning and memory in the brain. These synaptic updates are driven by local neuronal activity and global reward signals, and they in turn modify the dynamics of neural circuits. Advances in neuroscience have made it possible to record neuronal activity at an unprecedented scale (Steinmetz et al., 2018; Vanwalleghem et al., 2018; Zhang et al., 2023), and connectome data for various organisms is becoming increasingly available (Bentley et al., 2016; Hildebrand et al., 2017; Scheffer et al., 2020). However, the inaccessibility of direct large-scale recordings of synaptic dynamics leaves the identification of biological learning rules an open challenge. Existing neuroscience literature (Abbott & Nelson, 2000; Morrison et al., 2008) suggests that synaptic changes ($\Delta w$) are functions of local variables such as presynaptic activity ($x$), postsynaptic activity ($y$), and current synaptic weight ($w$), as well as a global reward signal ($r$). Uncovering the specific form of this function in different brain circuits promises deep biological understanding and holds practical significance for inspiring more biologically plausible learning algorithms for AI, in particular with neuromorphic implementations (Zenke & Neftci, 2021).

In this paper, we bridge computational neuroscience and deep learning to introduce a general framework for inferring synaptic plasticity rules. Our method optimizes plasticity rules to fit both neural and behavioral data, thereby shedding light on the mechanisms governing synaptic changes in bio-

logical systems. Our approach employs interpretable models of plasticity, parameterized to enable a direct comparison with existing biological theories.

In summary, our key contributions are:

- Introduction of a versatile framework for inferring synaptic plasticity rules from experimental data, accompanied by an open-source, extensible implementation.[1]

- Demonstration of a gradient descent based approach for recovering plasticity rule parameters solely from observed neural activity or probabilistic behavior, along with a characterization of rule recoverability.

- Application of our model to behavioral data from fruit flies, revealing an active forgetting mechanism in mushroom body circuits for decision making, learning, and memory.

## 2 RELATED WORK

Recent work has begun to address the question of understanding computational principles governing synaptic plasticity by developing data-driven frameworks to infer underlying plasticity rules from neural recordings. Lim et al. (2015) infer plasticity rules, as well as the neuronal transfer function, from firing rate distributions before and after repeated viewing of stimuli in a familiarity task. The authors make assumptions about the distribution of firing rates, as well as a first-order functional form of the learning rule. Chen et al. (2023) elaborate on this approach, fitting a plasticity rule by either a Gaussian process or Taylor expansion, either directly to the synaptic weight updates or indirectly through neural activity over the course of learning. Both approaches consider only the difference in synaptic weights before and after learning. In contrast, our approach fits neural firing rate *trajectories* over the course of learning and can be adapted to fit any parameterized plasticity rule.

Other work infers learning rules based on behavior instead. Ashwood et al. (2020) uses a Bayesian framework to fit parameters of learning rules in a rodent decision-making task. The authors explicitly optimize the weight trajectory in addition to parameters of the learning rules, requiring an approximation to the posterior of the weights. Our approach directly optimizes the match between the model and either neural or behavioral data, as defined by a pre-determined loss function. Interestingly, despite this indirect optimization, we see matching in the weight trajectories as well. Rajagopalan et al. (2023) fit plasticity rules in the same fly decision-making task we consider here. They assumed that the learning rule depended only on presynaptic activity and reward, which recasts the problem as logistic regression and permits easy optimization. Our approach allows us to account for arbitrary dependencies, such as on postsynaptic activities and synaptic weight values, and we thereby identify a weight decay term that leads to active forgetting.

Previous work has also considered inferring plasticity rules directly from spiking data (Stevenson & Koerding, 2011; Robinson et al., 2016; Linderman et al., 2014; Wei & Stevenson, 2021). Due to the gradient-based nature of our optimization technique, our proposed approach can only account for such data by converting spike trains to a rate-based representation by smoothing.

Alternatively, meta-learning techniques (Thrun & Pratt, 2012) can be used to discover synaptic plasticity rules optimized for specific computational tasks (Tyulmankov et al., 2022; Najarro & Risi, 2020; Confavreux et al., 2020; Bengio et al., 1990). The plasticity rules are represented as parameterized functions of pre- and post-synaptic activity and optimized through gradient descent or evolutionary algorithms to produce a desired network output. However, the task may not be well-defined in biological scenarios, and the network's computation may not be known a priori. Our method obviates the need for specifying the task, directly inferring plasticity rules from recorded neural activity or behavioral trajectories.

Finally, Nayebi et al. (2020) do not fit parameters of a learning rule at all, but use a classifier to distinguish among four classes of learning rules based on various statistics (e.g. mean, variance) of network observables (e.g. activities, weights). Similarly, Portes et al. (2022) propose a metric for distinguishing between supervised and reinforcement learning algorithms based on changes in neural activity flow fields in a recurrent neural network.

---

[1]https://anonymous.4open.science/r/MetaLearnPlasticity-A3E3/

## 3 METHOD OVERVIEW

Our goal is to infer the synaptic plasticity function by examining neural activity or behavioral trajectories in a learning organism. Specifically, we aim to find a function that prescribes changes in synaptic weights based on relevant biological variables. For simplicity, we consider plasticity localized to a single layer of a neural network

$$\boldsymbol{y}^t = \text{sigmoid}\left(W^t \boldsymbol{x}^t\right), \tag{1}$$

where $\boldsymbol{x}$ is the input to the plastic layer (the stimulus), $\boldsymbol{y}$ is the resulting output activity, and $t$ is time. The synaptic weight matrix $W^t$ is updated at each time step based on a biologically plausible plasticity function $g_\theta$, *i.e.*:

$$\Delta w_{ij}^t = g_\theta(x_j^t, y_i^t, w_{ij}^t, r^t), \tag{2}$$

where $\theta$ parameterizes the function, $x_j^t$ is the presynaptic neural activity, $y_i^t$ the postsynaptic activity, $w_{ij}^t$ the current synaptic weight between neurons $i$ and $j$, and $r^t$ a global reward signal that influences all synaptic connections. We further define a *readout* function $f$ that models observable variables $M = (\boldsymbol{m}^1, \ldots, \boldsymbol{m}^T)$ of the network output $\boldsymbol{y}$, *i.e.*, $\boldsymbol{m}^t = f(\boldsymbol{y}^t)$. Those readouts model experimental observations $O = (\boldsymbol{o}^1, \ldots, \boldsymbol{o}^T)$. In the context of neural activity fitting, the readout is a subset of $\boldsymbol{y}$, whereas for behavioral models the readout aggregates $\boldsymbol{y}$ to yield the probability of a specific action. We will introduce our specific choices for readout functions in the following sections.

The loss function $\mathcal{L}$ aggregates the differences (defined by a function $\ell$) between the model's readout $M$ and the experimentally observed output $O$ across all time steps, *i.e.*:

$$\mathcal{L}(M, O; \theta) = \sum_{t=1}^{T} \ell\left(\boldsymbol{m}^t, \boldsymbol{o}^t; \theta\right). \tag{3}$$

We can now use gradient descent to find plasticity parameters $\theta$, such that the loss between the model readout and observed values is minimized:

$$\theta \leftarrow \theta - \eta \nabla_\theta \mathcal{L}(M, O; \theta). \tag{4}$$

### 3.1 IMPLEMENTATION

In experimental contexts, the observed behavioral or neural trajectories ($O$) can span extensive time scales, often consisting of thousands of data points. Computing gradients over such long trajectories is computationally demanding. Our framework is implemented in JAX (Bradbury et al., 2018) and designed to accommodate plasticity modeling across millions of synapses, anticipating future integration with large-scale connectomics datasets. Performance benchmarks reveal that our framework is capable of executing simulations with up to $10^6$ synapses and 1000-time step trajectories in less than an hour on an NVIDIA A100 GPU.

## 4 INFERRING PLASTICITY FROM MEASURED NEURAL ACTIVITY

To illustrate and test this approach on neural activity, we generate a synthetic trajectory of observed outputs, denoted by $O$, using Oja's rule for synaptic plasticity (Oja, 1982). The update rule for Oja's plasticity rule is given as follows:

$$\Delta w_{ij} = x_j y_i - y_j^2 w_{ij}. \tag{5}$$

In our evaluation, we use Oja's rule as a known *a priori* standard to validate the accuracy of our model in inferring synaptic plasticity (Figure 2A). Since Oja's rule does not involve reward, we did not include a reward signal in this benchmark experiment. Inspired by previous work (Confavreux et al., 2020), we parameterize the plasticity function $g_\theta$ through a truncated Taylor series expansion,

$$g_\theta^{\text{Taylor}} = \sum_{\alpha, \beta, \gamma = 0}^{2} \theta_{\alpha, \beta, \gamma} x_i^\alpha y_j^\beta w_{ij}^\gamma. \tag{6}$$

Here, the coefficients $\theta_{\alpha, \beta, \gamma}$ represent the model's learned parameters for the Taylor series expansion and were initialized i.i.d. from a normal distribution of mean 0 and variance $10^{-4}$. It is worth noting

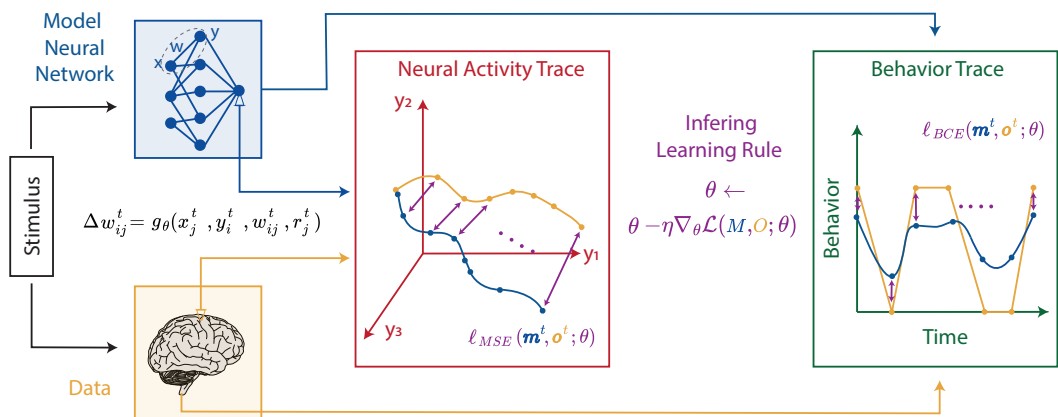

Figure 1: Schematic overview of the proposed method. Animal-derived time-series data (yellow) and a plasticity-regulated in silico model ($g_\theta$, blue) generate trajectories ($o^t$,$m^t$). A loss function quantifies trajectory mismatch to produce a gradient, enabling the inference of synaptic plasticity rules.

that Oja's rule can be represented within this family of plasticity functions by setting $\theta_{110} = 1$ and $\theta_{021} = -1$. For the following experiments, we set the readout function $f$ to be an activity readout of a subset of all neurons, *i.e.*, $M = f(\boldsymbol{y}) = (y_i | i \in S)$, where $S \subseteq [1, \ldots, n]$, and $n$ is the number of output neurons. We compute the loss as the Mean Squared Error (MSE) between the neural trajectories produced by both the Oja's rule-governed network and the Taylor series-based network,

$$\mathcal{L}_{\text{MSE}}(M, O; \theta) = \frac{1}{T} \sum_{t=1}^{T} ||\boldsymbol{o}^t - \boldsymbol{m}^t||^2. \tag{7}$$

### 4.1 RECOVERING OJA'S RULE

In our experimental setup, we use a single-layer neural network with 100 input neurons and 1000 output neurons to generate synthetic data with a known ground-truth plasticity rule. The output layer uses a sigmoid activation function for its differentiability and biologically plausible output range of [0,1]. The readout selects a subset of neurons. The network is simulated over 50 time steps, during which we calculate the neural activity of all output neurons at each time step. For numerical stability, we apply gradient clipping at 0.2 to circumvent exploding gradient issues. Despite fitting on neuronal activity, Figure 2B shows a clear decrease in the error of the synaptic weight trajectories (horizontal axis) over training epochs (vertical axis). Figure 2C shows that the training dynamics for the plasticity coefficients leads to a correct recovery of Oja's rule.

### 4.2 ROBUSTNESS TO NOISE AND SPARSITY OF RECORDING

To evaluate the robustness of our method, we assess how varying degrees of sparsity in recorded neural activity and additive noise affect the model's performance. We use the $\mathcal{R}^2$ metric to quantify the explained variance between the model-generated and observed weight trajectories post-training. Figure 2D illustrates the model's fit quality with increasing Gaussian noise levels and sparsity in the neuronal data. To simulate varying sparsity levels, denoted by $a$, we compute the loss using a fraction $a$ of the neurons from the observed trajectories. The remaining $(1 - a)$ fraction of neurons are modeled by incorporating random noise drawn from the same distribution as the observed neurons. Figure 2E delves into the noise sensitivity by exploring how different noise variances in the observed trajectories affect model performance. Figure 2F shows that our model retains a high degree of accuracy even when data from only 50% of the neurons is available, which is particularly advantageous given that sparse neural recordings are typical in experimental settings.

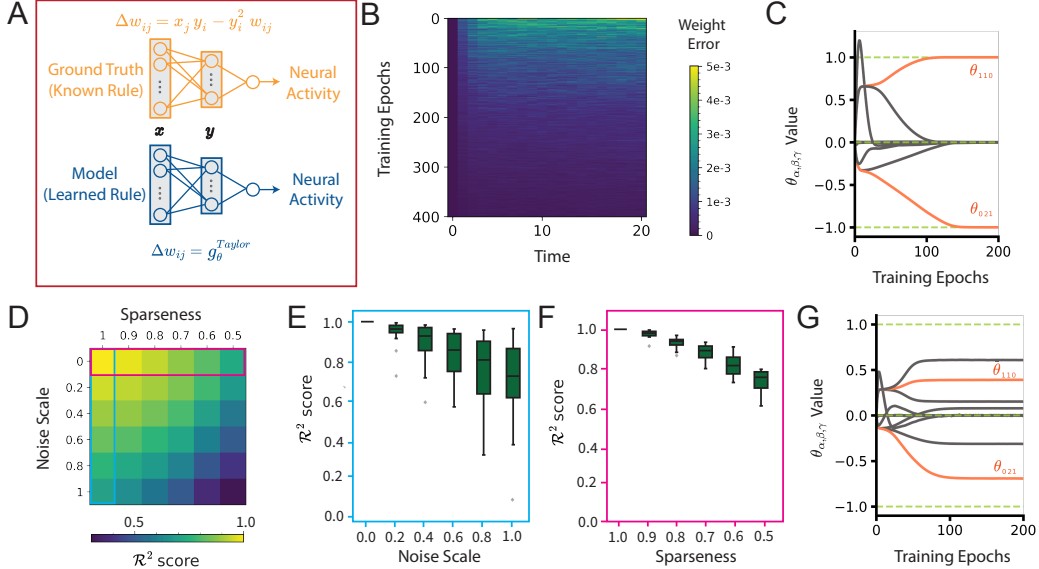

Figure 2: Recovery of Oja's plasticity rule from simulated neural activity. (A) Schematic of the models used to stimulate neural activity and infer plasticity. (B) Illustration of the weight trajectory distance between Oja's rule and the $g_\theta^{\text{Taylor}}$-updated models throughout training (shown as mean squared differences). (C) The evolution of $\theta$ during training. Coefficients $\theta_{110}$ and $\theta_{021}$, corresponding to Oja's rule values $(1, -1)$, are highlighted in orange. (D) $\mathcal{R}^2$ scores over weights, under varying noise and sparsity conditions in neural data. (E, F) Boxplots of distributions, across 50 seeds, corresponding to the first column (E) and row (F) in (D). (G) The evolution of learning rule coefficients over the course of training showing non-sparse $\theta$ recovery under high noise and sparsity conditions.

## 5 INFERRING PLASTICITY FROM MEASURED BEHAVIOR

In extending our model's capacity for inferring synaptic plasticity, we apply it to observed simulated behavioral data. In controlled experiments, animals are frequently presented with a series of stimuli, with their choices to accept or reject these stimuli resulting in rewards and subsequent synaptic changes. Here we model this plastic layer of neural connections and the resulting learned behavior. The output neurons of the model network generate probability scores, subsequently read out to sample a binary decision to either "accept" or "reject" a presented stimulus. Upon accepting a stimulus, a stochastic reward is administered, which then alters the plastic weights in the network.

The behavioral data available for training contains only these binary decisions. The model's synaptic weights are initialized randomly, mirroring real-world biological systems where initial synaptic configurations are typically unknown *a priori*. Our experimental setup is inspired by recent studies that have successfully mapped reward-based learning rules to observed behaviors in flies (Aso & Rubin, 2016; Rajagopalan et al., 2023). Furthermore, existing research indicates that the difference between received and expected rewards is instrumental in mediating synaptic plasticity (Schultz et al., 1997; Rajagopalan et al., 2023), and learning and forgetting happen on comparable timescales (Aso & Rubin, 2016).

For the synaptic updates, we therefore employ a reward-based plasticity rule (Figure 3A). The change in synaptic weight $\Delta w_{ij}$ is determined by the presynaptic input $x_j$, the existing synaptic weight $w$, and a globally shared reward signal $r$. We neglect hypothetical dependencies on $y_i$ because they are known to not impact reward learning in flies. We primarily emphasize results from simulating a covariance-based learning rule (Loewenstein & Seung, 2006) that was inferred from previous experiments (Rajagopalan et al., 2023). Here the reward signal is typically the deviation of the actual reward $R$ from its expected value $\mathbb{E}[R]$, calculated as a moving average over the last 10

trials

$$\Delta w_{ij} = x_j(R - \mathbb{E}[R]). \tag{8}$$

However, we also show that our method is also able to recover several biologically plausible alternatives.

For the model network, we employ a plasticity function parameterized through either a truncated Taylor series or a multilayer perceptron, *i.e.*,

$$g_\theta^{\text{Taylor}} = \sum_{\alpha,\beta,\gamma=0}^{2} \theta_{\alpha\beta\gamma} x_j^\alpha r^\beta w_{ij}^\gamma \quad \text{or} \quad g_\theta^{\text{MLP}} = \text{MLP}_\theta(x_j, r, w_{ij}). \tag{9}$$

In this behavioral paradigm, we utilize Binary Cross-Entropy (BCE) as the loss function to measure the discrepancy between the observed decisions and the model's output probabilities for accepting a stimulus:

$$\mathcal{L}_{\text{BCE}}(M, O; \theta) = -\frac{1}{T} \sum_{t=1}^{T} \left[ \boldsymbol{o}^t \log(\boldsymbol{m}^t) + (1 - \boldsymbol{o}^t) \log(1 - \boldsymbol{m}^t) \right]. \tag{10}$$

Note that this is also proportional to the negative log-likelihood function of the model. Parameters were initialized i.i.d. from a normal distribution of mean $0$ and variance $10^{-4}$.

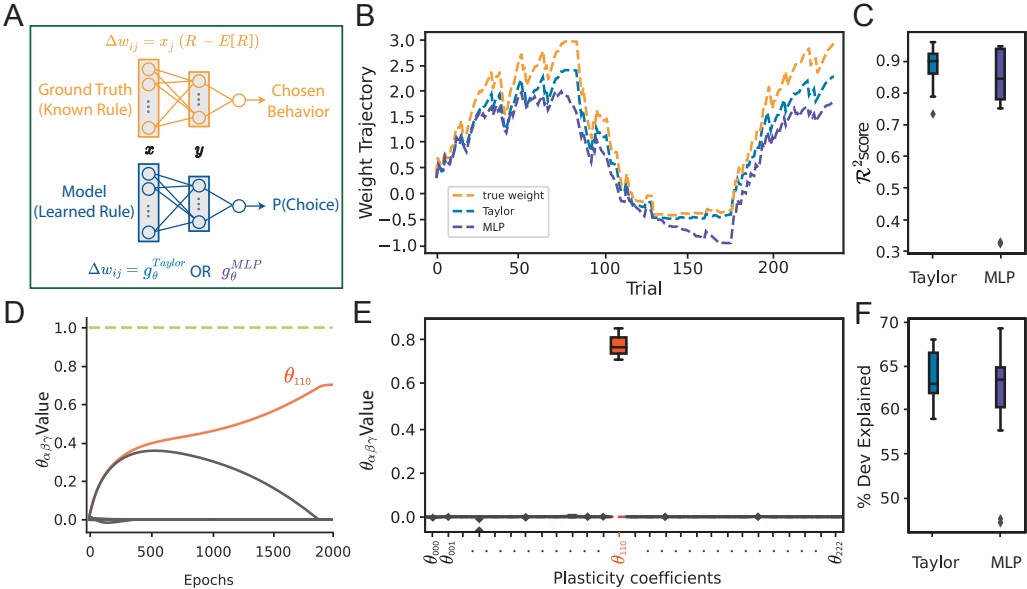

Figure 3: Recovery of a reward-based plasticity rule from simulated behavior. (A) Schematic of the models used to stimulate behavior and infer plasticity rules. (B) The evolution of the weight of a single synapse, trained with $g_\theta^{\text{Taylor}}$ and $g_\theta^{\text{MLP}}$, compared against weight from a known reward-based update rule. (C) $\mathcal{R}^2$ distributions across 20 seeds, corresponding to varied weight initializations and stimulus encodings. (D) The evolution of $\theta$ during training, with $\theta_{110}$, corresponding to ground truth rule (value = 1), highlighted in orange. (E) Distribution of final inferred $\theta$ values across seeds, showing accurate identification of the relevant term from the ground truth learning rule. (F) The goodness of fit between ground truth behavior and model predictions plotted as the percent deviance explained.

## 5.1 RECOVERING REWARD-BASED PLASTICITY FROM BEHAVIOR

The neural network model consists of three layers: an input layer with 100 neurons, an output layer containing 1000 neurons, and a single readout neuron that pools the output neurons to produce the probability of accepting a given stimulus. Upon choosing a stimulus, a probabilistic binary reward is delivered based on the choice. To encourage weight dynamics, the reward contingencies change

every 80 trials. Given the discrete nature of the observed behavior and the continuous output of the model, we employ the percent deviance explained as a performance metric. The percent deviance explained measures the model's ability to account for the variability in observed binary choices compared to a null model that assumes no plasticity. It represents the reduction in deviance relative to this null model, expressed as a percentage. Higher values are directly associated with greater log-likelihoods, signifying a superior fit to the observed data.

Figure 3B presents the weight dynamics of three models: a multilayer perceptron (MLP), a Taylor series approximation, and the ground-truth synaptic update mechanism. In the case of the MLP, we tested various architectures and highlight results for a 3-10-1 neuron topology. Our evaluation metrics—high $\mathcal{R}^2$ values for both synaptic weights and neural activity—affirm the robustness of our models in capturing the observed data (Figure 3C). The method accurately discerns the plasticity coefficient of the ground truth rule (Figures 3D,E), albeit with a reduced magnitude. The model also does a good job at explaining the observed behavior (Figure 3F).

Additionally, Table 1 summarizes the recoverability of various reward-based plasticity rules for both MLP and Taylor series frameworks, with results averaged over 10 random seeds. It's important to note that solely reward-based rules (without $\mathbb{E}[R]$ or $w$) are strictly potentiating, as they lack the capacity for bidirectional plasticity. This unidirectional potentiation ultimately results in the saturation of the sigmoidal non-linearity. Therefore, it is possible to simultaneously observe high $\mathcal{R}^2$ values for neural activities with low $\mathcal{R}^2$ values for weight trajectories.

Table 1: Evaluation of different reward-based plasticity rules: $\mathcal{R}^2$ scores for weight and activity trajectories, and percentage of deviance explained for behavior.

| Plasticity Rule | $\mathcal{R}^2$ (Weights) | | $\mathcal{R}^2$ (Activity) | | % Deviance Explained | |
| --- | --- | --- | --- | --- | --- | --- |
| | $g_\theta^{\text{Taylor}}$ | $g_\theta^{\text{MLP}}$ | $g_\theta^{\text{Taylor}}$ | $g_\theta^{\text{MLP}}$ | $g_\theta^{\text{Taylor}}$ | $g_\theta^{\text{MLP}}$ |
| $xR$ | -5.95 | -5.84 | 0.68 | 0.78 | 87.50 | 90.71 |
| $x^2R - 0.05xw$ | 0.35 | -39.28 | 0.79 | 0.87 | 86.01 | 90.17 |
| $x(R - \mathbb{E}[R])$ | 0.77 | 0.76 | 0.96 | 0.97 | 63.60 | 66.34 |
| $x^2(R - \mathbb{E}[R])$ | 0.85 | 0.75 | 0.96 | 0.94 | 55.24 | 55.93 |
| $R - \mathbb{E}[R] - 0.05w$ | 0.68 | 0.65 | 0.79 | 0.83 | 38.62 | 39.63 |
| $R - \mathbb{E}[R] - 0.05xw$ | 0.68 | 0.52 | 0.87 | 0.79 | 40.73 | 39.19 |
| $x(R - \mathbb{E}[R]) - 0.05w$ | 0.93 | 0.91 | 0.96 | 0.92 | 42.43 | 44.43 |
| $x(R - \mathbb{E}[R]) - 0.05xw$ | 0.85 | 0.86 | 0.91 | 0.92 | 49.72 | 51.18 |
| $x^2(R - \mathbb{E}[R]) - 0.05w$ | 0.96 | 0.96 | 0.97 | 0.96 | 36.02 | 37.62 |
| $x^2(R - \mathbb{E}[R])^2 - 0.05w^2$ | 0.18 | 0.31 | 0.66 | 0.76 | 73.05 | 75.81 |

# 6 EXPERIMENTAL APPLICATION: INFERRING PLASTICITY IN THE FRUIT FLY

In extending our model to biological data, we explore its applicability to decision-making behavior in the fruit fly, *Drosophila melanogaster*. Recent research (Rajagopalan et al., 2023) employed logistic regression to infer learning rules governing synaptic plasticity in the mushroom body, the fly's neural center for learning and memory. However, logistic regression cannot be used to infer plasticity rules that incorporate recurrent temporal dependencies, such as those that depend on current synaptic weights. Our method offers a more general approach. Specifically, we apply our model to behavioral data obtained from flies engaged in a two-alternative choice task, as outlined in Figure 4A. This allows us to investigate two key questions concerning the influence of synaptic weight on the plasticity rules governing the mushroom body.

## 6.1 EXPERIMENTAL SETUP AND DETAILS

In the experimental setup, individual flies were placed in a symmetrical Y-arena where they are presented with a choice between two odor cues (Rajagopalan et al., 2023). Each trial started with the fly in an arm filled with clean air (Fig. 4A, left). The remaining two arms were randomly filled with two different odors, and the fly was free to navigate between the three arms. When the fly

enters the 'reward zone' at the end of an odorized arm, a choice was considered to have been made (Fig. 4A, right). Rewards were then dispensed probabilistically, based on the odor chosen. For model fitting, we used data from 18 flies, each subjected to a protocol that mirrors the trial and block structures in the simulated experiments presented previously. Over time, flies consistently showed a preference for the odor associated with a higher probability of reward, and this preference adapted to changes in the relative value of the options (Fig. 4B; example fly).

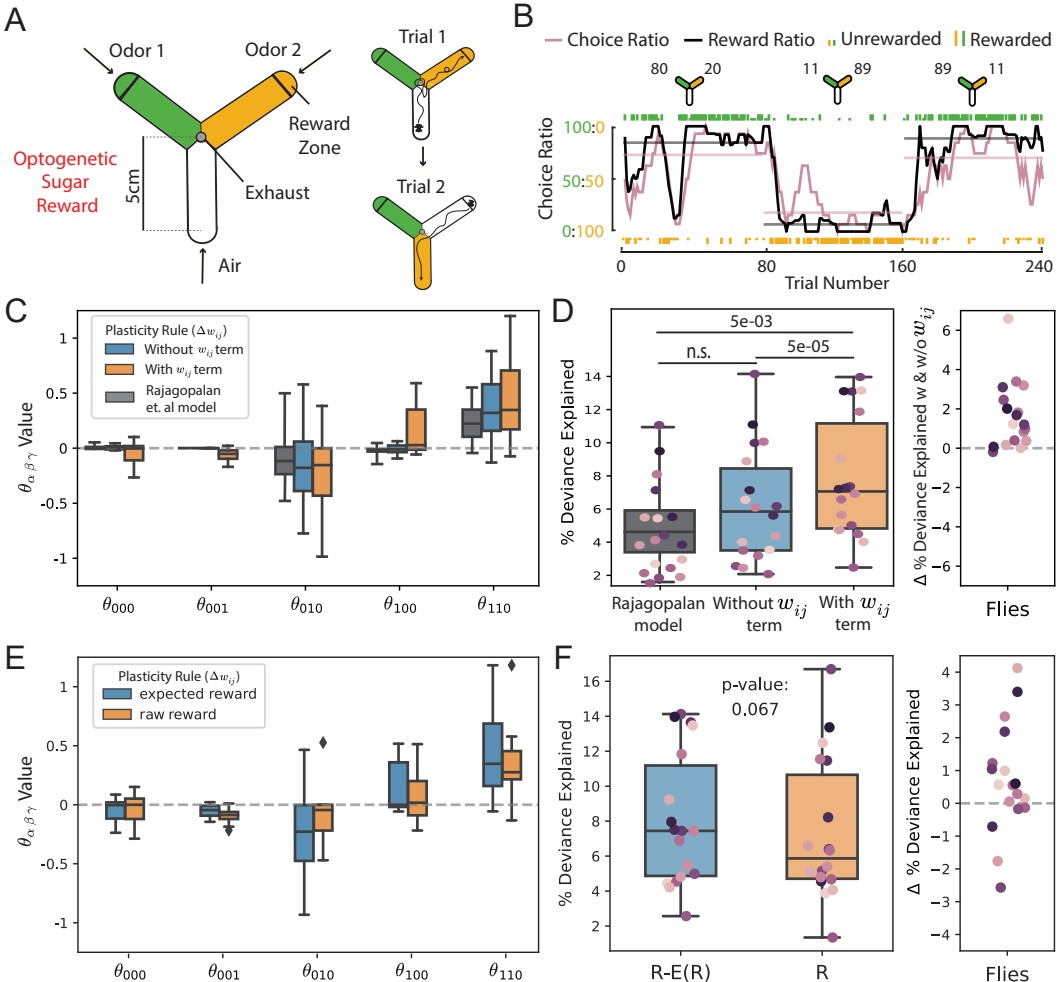

Figure 4: Inferring principles of plasticity in the fruit fly. (A) Schematic of the experimental setup used to study two-alternative choice behavior in flies. *Left* Design of arena showing how odor entry ports and location of the reward zones. *Right* Description of the trial structure showing two example trials. (B) The behavior of an example fly in the task. *Top* Schematics indicate the reward baiting probabilities for each odor in the three blocks. *Bottom* Individual odor choices are denoted by rasters, tall rasters - rewarded choices, short rasters - unrewarded choices. Curves show 10-trial averaged choice (red) and reward (black) ratio, and horizontal lines the corresponding averages over the 80-trial blocks. (C) Final inferred $\theta$ value distribution across 18 flies, comparing models with and without a $w_{ij}$ ($\theta_{001}$) term and the method from Rajagopalan et al. (2023). (D) *Left* Goodness of fit between fly behavior and model predictions plotted as the percent deviance explained (n = 18 flies). *Right* Change in the percent deviance explained calculated by subtracting percent deviance explained of model without a $w_{ij}$ ($\theta_{001}$) term from that of a model with a $w_{ij}$ ($\theta_{001}$) term. (E,F) Same as (C,D), except comparing models that do or don't incorporated reward expectation. Since these models include weight dependence, they cannot be fit using the method of Rajagopalan et al. (2023).

### 6.2 BIOLOGICAL QUESTION 1: DOES THE PLASTICITY RULE IN THE FRUIT FLY INCORPORATE A WEIGHT DEPENDENCE?

Existing behavioral studies in fruit flies have shown that these insects can forget learned associations between stimuli and rewards over time (Shuai et al., 2015; Aso & Rubin, 2016; Berry et al., 2018; Gkanias et al., 2022). One prevailing hypothesis attributes this forgetting to homeostatic adjustments in synaptic strength within the mushroom body (Davis & Zhong, 2017; Davis, 2023). However, earlier statistical approaches aimed at estimating the underlying synaptic plasticity rule present in the mushroom body were unable to account for recurrent dependencies such as synapse strength (Rajagopalan et al., 2023). Here we explore two types of plasticity rules: one based solely on reward and presynaptic activity, and another that also incorporates a term dependent on current synaptic weight. Both rule types allocate significant positive weights to a term representing the product of presynaptic activity and reward (Fig. 4C), gray. The model lacking dependence on current weight replicated the findings of Rajagopalan et al. (2023) (Fig. 4C) and is consistent with what generally known about the mushroom body (Modi et al., 2020). Our results indicate that the model with a weight-dependent term offers a better fit to observed fly behavior (Wilcoxon signed-rank test: p = $5 \cdot 10^{-5}$; Fig. 4D), whereas the model without it matched the performance reported in Rajagopalan et al. (2023). Intriguingly, our analysis additionally reveals that the inferred learning rule assigns a negative value to the weight-dependent term (Fig. 4C). This negative sign aligns with the hypothesis that a weight-dependent decay mechanism operates at these synapses. The model with the weight-dependent term also shows a greater coefficient for the purely presynaptic component (Fig. 4C).

### 6.3 BIOLOGICAL QUESTION 2: IS THE DIFFERENCE FROM THE EXPECTED REWARD BETTER THAN JUST REWARD?

The previous study used reward expectations to generate bidirectional synaptic plasticity (Rajagopalan et al., 2023). Our discovery of a negative weight-dependent component in the plasticity rule provides an alternate mechanism for this, raising the question of whether the neural circuit really needs to calculate reward expectation. Could a plasticity rule incorporating the product of presynaptic activity and absolute reward combine with a weight-dependent homeostatic term to approximate a plasticity rule that involves reward expectation? To answer this, we contrast two models: one using only the absolute reward and another using reward adjusted by its expectation, both complemented by weight-dependent terms. Our analyses show that adding a weight-dependent term enhances the predictive power of both models (Fig 4E,F). However, the model that also factors in reward expectations provides a superior fit for the majority of flies in the data set (Wilcoxon signed-rank test: p = 0.067; Fig 4F). These compelling preliminary findings reaffirm the utility of reward expectations for fly learning, and larger behavioral datasets could increase the statistical significance of the trend. Overall, our model-based inference approach, when applied to fly choice behavior, suggests that synaptic plasticity rules in the mushroom body of fruit flies are more intricate than previously understood. These insights could potentially inspire further experimental work to confirm the roles of weight-dependent homeostatic plasticity and reward expectation in shaping learning rules.

## 7 LIMITATIONS AND FUTURE WORK

Despite its strengths, our model has several limitations that offer avenues for future research. One such limitation is the lack of temporal dependencies in synaptic plasticity, neglecting biological phenomena like metaplasticity (Abraham, 2008). Extending our model to account for such temporal dynamics would increase its biological fidelity. Another issue is the model's "sloppiness" in the solution space; it fails to identify a unique, sparse solution even with extensive data. As neural recording technologies like Neuropixels (Steinmetz et al., 2021; 2018) and whole-brain imaging (Vanwalleghem et al., 2018) become more advanced, and connectome data for various organisms become increasingly available (Bentley et al., 2016; Scheffer et al., 2020; Hildebrand et al., 2017), there are exciting opportunities for validating and refining our approach. Incorporating these high-resolution, large-scale datasets into our model is a crucial next step. In particular, future work could focus on scaling our approach to work with large-scale neural recordings and connectomics, offering insights into the spatial organization of plasticity mechanisms. Additional considerations for future research include the challenges posed by unknown initial synaptic weights, the potential necessity for exact connectome information, and the adequacy of available behavioral data for model fitting.

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
