# OpenReview forum: "Model Based Inference of Synaptic Plasticity Rules"
_ICLR.cc/2024/Conference — Submitted to ICLR 2024_

### Official Review · Reviewer_s75f · 2023-10-28

**Soundness:** 3 good
**Presentation:** 3 good
**Contribution:** 2 fair
**Rating:** 6
**Confidence:** 3

**Summary:**

The paper proposes a gradient-based learning framework to infer biologically-plausible synaptic plasticity rules from experimental data. The method models the learning rule using polynomials of presynaptic, postsynaptic activities, current synaptic weight, and reward signal or with a multilayer perceptron. The proposed approach successfully recovers Oja's rule and plasticity in the fruit fly.

**Strengths:**

The paper has the following strenghts:

* It addresses a crucial question related to inferring synaptic plasticity from biological data, contributing to the field.

* The experiments conducted are intriguing and validates the proposed method.

* Limitations of the proposed algorithm are discussed at the end of the paper in detail.

* The availability of a documented and clear code enhances the reproducibility of this work.

**Weaknesses:**

I think the paper has several weaknesses. Please see the following list and the questions sections.

* The statement in the introduction regarding the biological plausibility of backpropagation may be too weak ("While the backpropagation ..., its biological plausibility remains a subject of debate."). It is widely accepted that backpropagation is biologically implausible.

* Regarding the following sentence in Section 3 "We further define a readout ... ,i.e., $\mathbf{m}^t = f(y^t).$", did you mean to write $\mathbf{m}^t = f(\mathbf{y}^t)$ ($\mathbf{y}^t$ with boldsymbol)?

* On page 4, "setting $\theta_{110} = 1$ and $\theta_{012} = -1$," the second term should be $\theta_{021}$ rather than $\theta_{012}$.

* The initialization of polynomial coefficient parameters is not clear, and it seems they are initialized close to zero according to  Figure 2. It would be valuable to explain how they were initialized.

* The paper models synaptic plasticity rules only for feedforward connections. It would be interesting to explore the impact of lateral connections (by adding additional terms in Equation 6). Have you experimented with such a setup?

* In page 7, the authors state that "In the case of the MLP, we tested various architectures and highlight results for a 3-10-1 neuron topology." What are the results for other various architectures? Putting them into the paper would also be valuable (as ablation studies).

* The hyperparameters for the experiments are missing? What is the learning rate, what is the optimizer, etc.?

* I do not see that much difference between the experiment presented in Section 4 and the experiment in (Confavreux et al., 2020) (Section 3.1) except the choice of optimization method. In your experimental setup, you also do not model the global reward. Therefore, I think it makes it more similar to the experiment in (Confavreux et al., 2020).

* Comparison to previous work is missing.

**Questions:**

* As a follow-up question related to my comment about the similarity with (Confavreux et al., 2020): (Confavreux et al., 2020) uses Covariance Matrix Adaptation Evolution Strategy (CMA-ES) as they state it has better scalibility with the network size compared to gradient based strategy. Why do you use gradient based strategy? What is the advantage?

* In the experimental setup in Section 5, if you do not neglect the hypothetical dependencies on $y_i$, would your framework correctly infer the learning rule, i.e., if you use $$g_\theta = \sum_{\alpha, \beta, \phi, \gamma} \theta_{\alpha, \beta, \phi, \gamma} x_j^\alpha r^\beta y_i^\phi w_{ij}^\gamma$$ would your model learn $\theta_{\alpha, \beta, \phi, \gamma} = 0 \quad \forall \phi \neq 0$?

**Details Of Ethics Concerns:**

No ethical concerns were identified in this paper.

---

> ### Author Response · Authors · 2023-11-17
> **Reply to Reviewer s75f (Part 1)**
>
> Dear Reviewer,
>
> We thank the Reviewer for their positive assessments of our work and suggestions for improvement. As detailed below, we will make many changes to address your suggestions.
>
> ## Replies to points brought up in *Weaknesses* section:
>
> **B1.** We agree. We will modify this sentence to be stronger.
>
> **B2.** Yes. Thank you for catching this typo. We will correct it.
>
> **B3.** Yes. Thank you for catching this typo. We will correct it.
>
> **B4.** The coefficients were randomly initialized i.i.d. from a normal distribution of zero mean and variance 1e-4. We will add this methodological detail to the paper.
>
> **B5.** We agree with the Reviewer that it would be interesting to extend our method to recurrent neural networks. However, we could perform this numerical experiment in our setup without changing Eq. 6. In particular, x would become the presynaptic neuron’s activity, y would become the postsynaptic neuron’s activity, and w would be the weight between these neurons. The bigger difference is that Eq. 1 would need to be updated to incorporate recurrent network dynamics. We have not experimented with recurrence in our model, and this change would entail significant changes to our code bank. We have therefore decided to postpone these experiments for future work.
>
> B6. We accepted the Reviewer’s suggestion and summarize additional experiments that varied the architecture of the MLP below. We found that the performance of the MLP architecture could be significantly improved by moderately increasing the size of the hidden layer. As suggested, we will include these simulations in our revised manuscript.
>
> ### Simulation information and results
> Plasticity MLP architecture: 3 → h_l → 1
>
> Network: 100 → 1000 → 1
>
> Ground truth plasticity rule: x.(R - E[R])
>
> Trained on simulated behavioral data
>
> | Number of hidden units (h_l) | R2 weights | R2 activity | % Deviance Explained |
> |------------------------------|------------|-------------|----------------------|
> | 10                           | 0.68       | 0.95        | 62.99                |
> | 50                           | 0.76       | 0.97        | 66.34                |
> | 100                          | 0.62       | 0.93        | 61.32                |
>
> **B7.** We will revise the manuscript to state these details. For the experiments, hyperparameters were fine tuned by grid search, with performance averaged across three random seeds. Hyperparameters included an L1 sparsity regularizer applied to the plasticity coefficients, with the optimal value found to be 5e-3, and a moving average window for calculating the expected reward, which showed minimal impact and was set at a value of 10. In terms of optimizers, our experiments included AdaBelief, Adam, and AdamW, using their default parameters for each. The results were comparable across these optimizers, and the findings presented in the paper are based on the use of Adam.
>
> **B8.** We agree with the reviewer, but we think that the choice of optimization method is highly significant. As explained in the Introduction and Related Work sections, our focus is on fitting a broad range of arbitrary rules, without being constrained by the specific computation they implement or their optimality. This approach reflects the flexible and expansive nature of our model. The method provided here provides an exciting opportunity to infer biological learning rules from real experimental data sets from biology, where the function of the circuit is a priori unknown. For instance, no one knows if the mushroom body is optimal for some specific task, but we were nevertheless able to estimate its learning rule from behavioral data in Figure 4.
>
> **B9.** Can the Reviewer please clarify what comparison they would like to see? We already conceptually compared our work to previous papers in the Related Work section. However, since we’re optimizing different objective functions than almost all of these previous works, apples-to-apples comparisons are usually impossible. In the setting of Figure 4, we could revise the text to compare our model’s performance to the previous model in Rajagopalan et al. Would this help, or does the Reviewer have something else in mind?

---

> ### Author Response · Authors · 2023-11-17
> **Reply to Reviewer s75f (Part 2)**
>
> ## Replies to points brought up in *Questions* section:
>
> **B1.** Our method is simpler to implement and appears to work perfectly well for our purposes. We therefore did not explore the CMA-ES approach. We note that Tyulmankov et al. 2022 also successfully used gradient descent to meta-learn plasticity rules, and our implementation more closely follows that paper.
>
> **B2.** This is a good suggestion. We did not explore this learning rule because it involves significantly more coefficients that must be fit, and hypothetical dependencies of the mushroom body’s learning rule on y can be excluded on biological grounds. We will mention this in our revision. We will also try fitting this more complex learning rule in our revision to see whether the coefficients can still be recovered.

---

> > ### Comment · Reviewer_s75f · 2023-11-18
> >
> > Dear authors,
> >
> > I would like to thank you for your detailed answer. Regarding your following question:
> >
> > >  Can the Reviewer please clarify what comparison they would like to see? ...  In the setting of Figure 4, we could revise the text to compare our model’s performance to the previous model in Rajagopalan et al. Would this help, or does the Reviewer have something else in mind?
> >
> > I believe this would be a valuable addition (if it is plausible in the given limited time). Other than that, I am happy to move my Soundness and Presentation scores by 1 given these details will be added to the manuscript.

---

> > > ### Author Response · Authors · 2023-11-20
> > > **Revised manuscript incorporating changes to address weaknesses**
> > >
> > > We thank the reviewer for re-evaluating their presentation and soundness scores and for clarifying the comparisons they feel would further aid the paper. We have now revised our manuscript document to act on their suggestions. In particular, we have **implemented changes to fully address the “Weaknesses” bullets 1-4 and 9**. We have also started to address the sixth “Weaknesses” bullet by updating Table 1 with numbers from the new MLP architecture that showed better performance. We thank the reviewer for this helpful suggestion. We will finish making the changes promised in “Weaknesses” bullets 6-7 and “Questions” bullet 2 when we have time after the rebuttal period.
> > >
> > > **The changes we already made to address bullet 9** are most significant, so we summarize them briefly here. We have now added or edited multiple sentences in Section 6.2 to more explicitly compare our model with the earlier model presented by Rajagopalan et al. We have additionally added an explicit comparison in Figs. 4C-D, which has been updated to include the plasticity rule as inferred using Rajagopalan et al.’s method. As expected, excluding weight decay in our model results in the same plasticity rule recovered by Rajagopalan et al. (Fig. 4C), but our complete model is able to outperform it (Fig. 4D).
> > >
> > > With these implemented and planned changes, we believe that we will have been able to address all of the reviewer’s major concerns. Please let us know if there are any lingering issues that prevent the reviewer from considering updating their overall rating.

---

> > > > ### Comment · Reviewer_s75f · 2023-11-22
> > > >
> > > > Dear authors,
> > > >
> > > > I would like to thank you for the revised manuscript and the authors' commitment to providing further clarifications. I upgraded my score by one.

---

### Official Review · Reviewer_qMGi · 2023-10-28

**Soundness:** 3 good
**Presentation:** 3 good
**Contribution:** 3 good
**Rating:** 5
**Confidence:** 4

**Summary:**

This paper infers synaptic plasticity rules from experimental data on neural activity or behavioral trajectories by parameterizing the plasticity function to provide theoretical interpretability and facilitate gradient-based optimization.
They use Taylor series expansions or multilayer perceptrons to approximate plasticity rules and adjust parameters via gradient descent over entire trajectories to match observed neural activity or behavioral data closely.
They also learn intricate rules that induce long nonlinear time-dependencies, such as those incorporating postsynaptic activity and current synaptic weights, and validate through simulations, accurately recovering established rules, like Oja's, as well as more complex hypothetical rules incorporating reward-modulated terms.

**Strengths:**

The framework's ability to parameterize the plasticity function provides a balance between theoretical interpretability and practical optimization.

The use of established mathematical techniques, like Taylor series expansions and multilayer perceptrons, lends credibility to the approach.

The method's versatility is evident in its ability to learn both established rules like Oja’s and more complex, hypothetical ones.

Real-world application on Drosophila data and the discovery of an active forgetting component showcase the practical relevance and potential breakthroughs the framework can offer.

**Weaknesses:**

The paper is really interesting, but I have some key concerns. First and most importantly, I think this paper might not be a very good fit for the conference and might be better suited for a neuroscience journal/conference. For example, the paper tries to recover Oja's rule from experimental data. Though it is a good exercise, I feel it would be interesting if this method could be used to model a more generalized learning form and show the performance of using that learning method in small networks - suppose a Hopfield network with the new plasticity. Essentially, improves on top of Oja's method using the added information and makes it more biologically plausible.

**Questions:**

1. How were the plasticity rules shown in Table 1 derived? Were they ad-hoc variations of Oja's rule? It would be interesting if you could use some symbolic regression to find the optimal plasticity rule.

2. Can you explain what you mean by "fits neural firing rate trajectories over the course of learning" in Page 2?

3. You mentioned "Performance benchmarks reveal that our framework
is capable of executing simulations with up to 106 synapses and 1000-time step trajectories in less
than an hour on an NVIDIA A100 GPU." - however, from what I understand, the experiments seem to have been done on a much smaller number of synapses. Can you clarify?

4. I could not fully understand the results in Section 4.2. It would be great if the authors  could add more details on "The remaining (1 − a) fraction of neurons are modeled by incorporating random noise drawn from the same distribution as the observed neurons."

---

> ### Author Response · Authors · 2023-11-17
> **Response to Reviewer qMGi (Part 1)**
>
> Dear Reviewer,
>
> Thank you for your feedback and your recognition of the interesting aspects of our paper. Regarding your concerns:
>
> 1. **Machine learning versus neuroscience**:
> We think that a machine learning conference proceeding is a better fit for our work than a neuroscience paper for two main reasons. First, typical neuroscience papers ask an interesting scientific question and then answer it through some combination of method development, experimentation, data analysis, and modeling. Answering a scientific question is not the point of this work. Instead, we aim to introduce a powerful general methodology and illustrate it with proof-of-concept applications. When neuroscientists write a paper like this, they usually publish it as a methods paper, and ours is a machine learning method that fits a machine learning conference perfectly. It is common for papers like ours to be published in machine learning conferences, as evidenced by the many machine learning conference papers that we cite in our paper. Second, our paper introduces a methodology for understanding learning in neural networks that, while demonstrated within a biological context, has broader applications. The principles and techniques we outline are relevant not only to neuroscience but also to the field of artificial neural networks (ANNs) and deep learning. For instance, our method could be used to learn a simple parameterized learning rule that approximates the learning trajectories of a more computationally intensive method. This interdisciplinary relevance makes our work a good fit for the conference, as it bridges the gap between biological neural mechanisms and computational models in machine learning.
> In summary, we believe that our work contributes significantly to the understanding of learning in neural networks, with implications that span across neuroscience and artificial intelligence. We are open to further discussions and clarifications on how this aligns well with the conference scope.
>
> 2. **Choice of Oja's rule**:
> The use of Oja's rule in our study is primarily for validation purposes. It serves as an arbitrary, yet well established, benchmark to demonstrate the efficacy of our methodology. Our choice of Oja's rule was driven by its recognizability and established theoretical foundations, making it a suitable candidate for ground truth comparison. However, it's important to note that our methodology is not restricted to Oja's rule; it can be applied to infer a wide range of synaptic plasticity rules. For example, the alternatives in Table 1 were not derived from any theoretical objective, they were merely many examples of learning rules that could be used to generate neural trajectories, and our method was consistently able to recover them. We do not think that it is necessary to use symbolic regression, or any other method, to find the optimal plasticity rule. Although it is interesting to relate the learning rules of single neurons to the output behavior of the whole system, that is not the topic of our paper. Instead, our focus is on fitting a broad range of arbitrary rules, without being constrained by the specific computation they implement or their optimality. This approach reflects the flexible and expansive nature of our model.
>
> 3. **Generalized learning forms and network performance**: Your suggestion to model a more generalized learning form and benchmark its performance in networks like a Hopfield network is a good idea. However, that is not the topic of this paper, which you already recognize as “really interesting.” Our current focus is on developing and validating a method for inferring learning rules, rather than on enhancing or modifying specific learning methods like Oja's rule. That our paper suggests another interesting computational problem to the reviewer lends further credence to our view that a machine learning conference proceeding fits our work well.

---

> > ### Author Response · Authors · 2023-11-17
> > **Response to Reviewer qMGi (Part 2)**
> >
> > 4. **Additional questions**
> >
> >     - Q2. The method minimizes the difference between the measured (experimental or simulated) neural trajectories and those generated by the model network. We will edit the text to make this clearer for future readers.
> >     - Q3. It’s already possible to record a hundred thousand neurons in larval zebrafish, and, given emerging trends in neuroscience, there is potential for future technologies to enable the recording of millions of neurons. This activity will stem from orders of magnitude more synapses. Although our current datasets do not encompass this scale, our synthetic data experiments show consistent results for both large and small networks. Crucially, our framework is designed to efficiently handle data from millions of neurons, preparing us for when such extensive recordings become available. We therefore think that it is important to quantitatively benchmark the performance of our approach.
> >     - Q4. We recognize the importance of this detail and will include a clarification in our revised paper. In brief, for the unobserved (1 − a) fraction of neurons in our model, we incorporate random noise drawn from the same distribution as the observed neurons (the initial distribution). This is because the exact values of these unobserved neurons are unknown, but they nevertheless have a big impact on the activity level of downstream neurons. We assume knowledge of their overall activity distribution to incorporate this effect.

---

> > > ### Author Response · Authors · 2023-11-20
> > > **Revised manuscript, and request for additional feedback**
> > >
> > > We want to alert the reviewer that we have posted an updated manuscript that concretely addresses some key concerns that have come up in conversation with Reviewers 2 and 4. We believe that the revision we have proposed above will also address all of this reviewer’s major questions and concerns. Please let us know if there are any lingering issues that prevent the reviewer from considering updating their overall rating, as we would like to address them in the rebuttal period.

---

> ### Comment · Reviewer_qMGi · 2023-11-22
> **Thank you authors**
>
> I would like to thank the authors for their feedback. I like the paper. However, I still don't fully understand the significance. The authors mentioned, "our method could be used to learn a simple parameterized learning rule that approximates the learning trajectories of a more computationally intensive method" and "our methodology is not restricted to Oja's rule; it can be applied to infer a wide range of synaptic plasticity rules" - I believe it is necessary to show these cases in this paper. Otherwise, I am not very convinced regarding the substantiality of the novelty of the current manuscript.
>
> For this reason, I will stick to my original score. Thank you, and good luck!

---

> > ### Author Response · Authors · 2023-11-22
> > **Thank you and very quick follow-up point**
> >
> > Thank you for considering an update to your score. We respect your decision. Nevertheless, we do want to remark that we benchmarked our method on 11 different learning rules between Figure 2 and Table 1. We just wanted to make sure that was clear before we rest our case, as we already showed that our method "can be applied to infer a wide range of synaptic plasticity rules." You are correct that we did not yet use our method to approximate a more computationally intensive method. In any case, thank you for the time you have given to reviewing our paper.

---

### Official Review · Reviewer_QuAL · 2023-10-31

**Soundness:** 2 fair
**Presentation:** 3 good
**Contribution:** 2 fair
**Rating:** 5
**Confidence:** 3

**Summary:**

The paper proposes to learn parametrized synaptic plasticity rules in recurrent rate networks by optimizing on loss functions defined on the neural trajectories and/or behavioural data. The method is validated using Oja's rule, and also applied to neural and behavioural data from the fruit fly. In the latter experiment, the method identifies a weight-dependent term that accounts for better predictions of fly behaviour.

**Strengths:**

1. The idea to directly fit plasticity rules to neural or behavioural data, rather than to functional tasks, to get biologically plausible rules is interesting.

2. The experiments with the fly data are also interesting. In particular, the careful exploration to test different hypothesis about the synaptic mechanism underlying fly choice behaviour led to interpretable results on which plasticity rules, and in particular, which terms in the rule could modulate forgetting learned associations, and bidirectional plasticity.

3. The paper is clearly written, concepts and related work are well-explained and overall, quite easy to follow.

**Weaknesses:**

1. The fact that the method is restricted to rate networks precludes interesting synaptic rules from spiking networks (e.g. spike-timing dependent plasticity) that could explain a wider a range of plasticity mechanisms, and thus limits the set of interesting rules that could be inferred using this method.

2. I am skeptical of the generalisability of this method, since the discovered rules would strongly depend on the choice of loss function for the neural / behavioural data. While the experiments in section 4.2 with robustness to noise and sparsity work in the idealised setting, the robustness is demonstrated with Gaussian noise, which the MSE loss is by definition capable of ignoring, in the limit of infinite data. However, with real-world neural/behavioural data, we cannot always pinpoint the exact distribution of noise, and it would be practically quite difficult to construct a loss function that is robust to unknown noise. I am thus skeptical that the method would generalise to arbatrarily complex neural behaviours or trajectories, and that the inferred plasticity rules in these settings would be biologically plausible.

3. Section 7 mentions that "[a]nother issue is the model's "sloppiness" in the solution space; it fails to identify a unique sparse solution even with extensive data". If this is the case with the current experiments in sections 4-6, how does this affect the interpretation of the results and how much can we rely on the experimental predictions about plasticity mechanisms? For instance, which particular experiment/solution forms the basis for the conclusions about the decay mechanism in 6.1 and bidirectional plasticity in 6.2?

4. The experiments in section 6, appear to add terms to the parametrized rule based on previously hypothesised plasticity mechanisms in the fruit fly. Although, this was an effective method of exploration in this particular instance, how would this generalise to neural / behavioural datasets for which such hypotheses do not exist, or are unsatisfactory? If instead, we could perform rule inference using this method using MLPs or some other form of parametrization, how would we interpret the resulting rules?

Taken together, it seems that although this method provides more biologically plausible rules that plasticity rule inference via optimizing for network function (Tyulmankov et al 2022, Confavreux et al 2020, Lindsey and Litwin-Kumar 2020, etc.), it swaps one set of loss functions for another, but still shares many of the same problems of generalisability and interpretation.

**Questions:**

1. How generalisable is the method, when in its current form, it cannot be applied to spiking networks?
2. How generalisable is the method, when constructing a loss function on neural activity / behaviour with unknown noise sources?
3. How interpretable / reliable are the inferred plasticity mechanisms when the model fit is "sloppy" with limited data?
4. How generalisable is the inference mechanism when we do not have satisfactory a priori hypotheses about the underlying plasticity mechanism?

---

> ### Author Response · Authors · 2023-11-17
> **Response to Reviewer QuAL (Part 1)**
>
> Dear Reviewer,
>
> Thank you for your comments on our work. We address each of the suggested weaknesses/questions below. To summarize, while we agree with the highlighted limitations due to robustness (noise/sloppiness), interpretability, and lack of prior hypotheses, these concerns are applicable to any statistical model-fitting setup. Some of these have standard solutions, but in general they are ubiquitous problems that are outside the scope of our work. We also point out that our method indeed can account for spiking models, and the plasticity rules that we fit are, by definition, biologically plausible.
>
> **Spiking neural network models**
>
> Although our approach uses rate networks as the underlying neural network model, our approach is not limited to rate networks. In the case of stochastic spiking models, we demonstrate an analogous scenario by fitting fly behavioral data with a probabilistic model, maximizing the log-likelihood of the model’s output given the data. This can easily be applied to fitting spiking (e.g. Poisson) neural trajectories rather than stochastic behavioral trajectories.
>
> Alternatively, we can think of rate-based neural networks as an abstraction over spiking networks. Given spiking data, we can generate a rate code by smoothing e.g. with a Gaussian kernel, which we can fit immediately with a rate model. Furthermore, STDP rules can be mapped onto Hebbian plasticity in rate networks [1], so rate networks do not preclude STDP as a potential plasticity mechanism (although we do concede that the precise timing of the spikes is lost).
>
> Our approach can also be easily modified to account for spiking data without any preprocessing, using a spiking model. As presented in the current work, the only limitation is the optimization algorithm. If the spiking model is an Hodgkin-Huxley type model with differentiable spike waveforms, we can use our method out-of-the-box – the only limitation is compute power, since these models have extremely detailed dynamics and many CPU cycles are required to simulate a single spike. If the spiking model is comprised of binary neurons with a Heaviside step function nonlinearity, optimization techniques such as surrogate gradients can be used. In the case of non-differentiable models, e.g. integrate-and-fire neurons, we can use evolutionary algorithms.
>
> **Unknown noise sources and biological plausibility**
>
> We agree that the noise model and loss function would impact the results of the model fitting. However, this is a problem that is true for any statistical modeling, and therefore much more general than the scope of our work. The Gaussian noise assumption is a common one and is used as the noise model ubiquitously. More importantly, Gaussian noise is also the most likely, assuming the Central Limit Theorem holds, and therefore it is a valid assumption. Similarly, mean-squared error is the standard loss function for many statistical models of real-valued data, and cross-entropy for models of binary data.
>
> If, however, there is strong prior belief that the noise is non-Gaussian, e.g. if there are correlations, bias, skew, or heavy tails in the noise model, this can be factored into the loss function appropriately. For completeness, we also note that after our initial submission, a preprint was released that directly tackles this problem by using a generative adversarial network to fit neural trajectories rather than an explicit loss function [2]. Although this approach is more general from the perspective of the noise model, it is both more demanding computationally, and a more difficult loss  landscape to optimize due to the use of a GAN. Although we were unable to include this work in the initial submission, we will make this comparison in our final version.
>
> Finally, the concern about the biological plausibility of the inferred plasticity rules in complex neural behaviors or trajectories is an important one. We would like to emphasize that our model infers plasticity rules using only local information, such as presynaptic and postsynaptic activity, current synaptic weight, and global dopaminergic reward signals. This, by definition, constrains the plasticity rules to be biologically plausible.

---

> ### Author Response · Authors · 2023-11-17
> **Response to Reviewer QuAL (Part 2)**
>
> **Sloppiness in fit and interpretability/reliability**
>
> Although the sloppiness of fit is a real problem in our models, we again think that it is a generic problem in statistical model fitting rather than a limitation in our specific approach. Interpretability depends on the functional form of the model (or equivalently, in the case of probabilistic graphical models, its graph structure). We demonstrate our method with an interpretable plasticity model in the form of a Taylor expansion, as well as a less interpretable one in the form of an MLP. The reliability of the method, as with all statistical models, depends on the quality and the quantity of the data. Standard statistical techniques can be used in these cases – confidence intervals, bootstrapping, cross-validation, etc. Some hypotheses, however, by definition cannot be distinguished with a particular dataset.
>
> We would also like to highlight the fact that the sloppiness occurs in the specific parameterization of our plasticity rule, rather than in a degeneracy of the rule itself (unlike the results in [2], which, again, we will discuss in the final version of our submission). We find a unique plasticity function, i.e., the shape of the input-output curve on a graph, but not a unique parametrization of that function.As a trivial example, the functions $y = x$ and $y = x^2$ are indistinguishable if the only values of $x$ which we sample are 0 and 1, so if the ground truth is $y = x + x^2$, we can fit any function $ax + bx^2$ with $a + b = 1$.
>
> Nevertheless, regardless of the parametrization of the plasticity rule, our method is informative with regard to the inputs that are relevant for a plasticity update, e.g. whether pre- or post-synaptic activity plays a role, or both; or, as we demonstrate in Drosophila, whether a weight decay is present. Following the adage, “All models are wrong but some are useful,” we believe that our model is useful from this perspective.
>
> **No a priori hypotheses**
>
> The problem of hypothesis generation is an important and difficult one. Any fitting experiment is limited to the functional family that is being fitted. Since the goal of our work is introducing and demonstrating a new and general computational methodology, not on establishing and interpreting a novel biological finding, we simply chose a hypothesis using prior knowledge established in other work. We demonstrate that our method is reliable for a given hypothesis class. Generating a hypothesis (i.e. the functional family as well as choosing the appropriate inputs) can be done through what has been referred to as "Box's loop" [3]. We iterate on hypotheses and do model validation for each one until we reach satisfactory convergence (or a deadline).
>
> In an extreme case, given sufficient data and computational power, we can simply consider a sufficiently broad hypothesis which includes all the possible inputs and allow the optimization routine to decide which ones should be used and in what way. If the functional family is an interpretable one, such as the Taylor series, we can gain mechanistic insights into the plasticity rule itself. For instance, consider
>
> $$
> g_\theta = \sum_{\alpha,\beta,\Phi,\gamma} \theta_{\alpha,\beta,\Phi,\gamma} x_j^\alpha r^\beta y_i^\Phi w_{ij}^\gamma
> $$
>
> We did not explore this learning rule because it involves significantly more coefficients that must be fit, and hypothetical dependencies of the mushroom body’s learning rule on y can be excluded on biological grounds. To address the reviewers concern, we will try fitting this more complex learning rule in our revision to see whether the coefficients can still be recovered.
>
> In a fully general approach, we can even consider a dynamic plasticity model (e.g. an RNN) that, given all possible inputs to a synapse, can learn history-dependent terms such as expected reward (which, this time, we manually included in our function). Although such a model (or even a simpler one such as the MLP) is hard to interpret, it still gives us an input-output relationship that we can use to make predictions. It also allows us to explore which inputs are important e.g. by including/excluding reward as one of the inputs into the MLP. Moreover, it gives us a well-defined functional form of a putative plasticity rule that we can perturb and analyze in ways unavailable in a biological system. Since the input space is low-dimensional, for example, we can visualize the function’s graph.
>
> [1] Sjostrom and Gerstner. Spike-timing dependent plasticity. Scholarpedia, http://www.scholarpedia.org/article/Spike-timing_dependent_plasticity#STDP_versus_Rate_based_learning_rules
>
> [2] Ramesh et al. Indistinguishable network dynamics can emerge from unalike plasticity rules. bioRxiv, https://www.biorxiv.org/content/10.1101/2023.11.01.565168v1.full.pdf
>
> [3] Blei. Build, Compute, Critique, Repeat: Data Analysis with Latent Variable Models. https://www.cs.columbia.edu/~blei/papers/Blei2014b.pdf

---

> ### Comment · Reviewer_QuAL · 2023-11-19
> **Thank you**
>
> I thank the authors for their detailed responses to my questions.
>
> Regarding the point on spiking neural network models, I agree that evolutionary algorithms or MLE estimation might indeed be a useful way to extend the framework spiking network models. However, I am still worried about the framework's generalizability, particularly when also accounting for non-Gaussian noise:
> > If, however, there is strong prior belief that the noise is non-Gaussian, e.g. if there are correlations, bias, skew, or heavy tails in the noise model, this can be factored into the loss function appropriately.
>
> Factoring these terms into the loss function might make MLE or evolutionary methods non-trivial to implement and train to convergence. In particular, while the assumption of Gaussian noise / central limit theorem might be ubiquitous, it is not uncommon to find correlated neural trajectories (Smith and Kohn 2005, Bartolo et al 2020, Valente et al 2021) or other shifts away from purely Gaussian noise.
> Given these difficulties, I am still skeptical about the generalizability of the framework.
>
> Regarding the sloppiness of fit:
> > We would also like to highlight the fact that the sloppiness occurs in the specific parameterization of our plasticity rule, rather than in a degeneracy of the rule itself
>
> I am confused by this -- could the authors elaborate on this point? Following from the example in the response, wouldn't the ambiguity in the values of $a$ and $b$ qualify as degeneracy? In any case, it would be useful to include this point in the discussion in the manuscript.
>
> Regarding a-priori hypothesis:
> > we will try fitting this more complex learning rule in our revision to see whether the coefficients can still be recovered.
>
> This would be a great addition. Comparisons/Discussion of Ramesh et al or Rajagopalan et al (as pointed out by reviewer s75f) would certainly be useful.
>
> I recognize that this may be difficult to achieve within the time-frame of discussion period. I have increased my score to a 5, on the understanding that these comparisons / discussions will be included in subsequent revisions of the paper if accepted.
>
> However, I would like to note that my concerns remain on the generalizability as well as limited novelty of this work.

---

> > ### Author Response · Authors · 2023-11-20
> > **Revised manuscript featuring direct comparisons and addressing concerns regarding generality and novelty (Part 1)**
> >
> > We thank the Reviewer for re-evaluating their score and for their valuable follow-up questions. We address the specific points below.
> >
> > We first want to alert the Reviewer that **we have uploaded a revised manuscript** that includes the comparison to Rajagopalan et al. In brief, the approach used by Rajagopalan et al. relied upon mapping the learning rule onto a logistic regression problem, and it was therefore unable to fit plasticity rules that involved recurrent statistical dependencies. Our method utilizes backpropagation through time and can accurately incorporate these dependencies. An example of this is the weight decay term, which we show to be of importance in Drosophila behavioral data.
> >
> > **We now provide a direct comparison** of our method to the one proposed by Rajagopalan et al. (see revised section 6.2 and updated Figs. 4C-D). We show that excluding the weight decay in our method results in the same plasticity rule recovered by Rajagopalan et al. We also modified the manuscript to explicitly show that the full model’s predictive performance exceeded that achievable by the method in Rajagopalan et al.  Overall, we find that our model and fitting method produce a better fit to the experimental data by allowing for a weight decay term that could not be fit by the method of Rajagopalan et al., and our revised manuscript makes this comparison explicit. There is insufficient time in the rebuttal period to implement the method of Ramesh et al. for numerical comparison. We will try to do this quantitative comparison if our paper is accepted. However, we remind the reviewer that Ramesh et al. is an unreviewed preprint that was posted after our submission, and we don’t yet know how feasible this quantitative comparison will actually be. At the very least, we will include a careful qualitative comparison if our paper is accepted.
> >
> > Second, we’re happy to **elaborate on our points regarding sloppiness.** Our example was meant to convey that the values of $a$ and $b$ were only impossible to resolve over a discrete input domain that approximates the data we were fitting ($x=0$ and $x=1$). In this case, although the functional forms are different, the input-output relationship at the points $x=0$ and $x=1$ are mathematically identical. In contrast, calculus shows that the Taylor series expansion of a continuously differentiable function is unique when defined over an open interval, which means that the plasticity function itself is non-degenerate when considered over all possible inputs (excepting the unlikely scenario in which biological plasticity rules are non-analytic functions). Nevertheless, sufficiently close to the points $x=0$ and $x=1$, a large amount of data may be necessary to estimate the correct coefficients from the data. Thus, although we added noise to the inputs in our model, the plasticity coefficients may not perfectly resolve the true values of $a$ and $b$. Following nomenclature from physics, we refer to this as "sloppiness" rather than "degeneracy," the latter referring to truly different functions that would lead to the same fit performance given infinite data. Please let us know if further clarification is required.
> >
> > **Finally, we respectfully disagree with the reviewer’s concerns on generality and novelty.**
> >
> > - We first note that our approach can already optimize an arbitrary objective function using backpropagation through time. Our method can thus implement MLE. Through the behavior examples, we already showed that our method works accurately for the cross-entropy loss, which is the log-likelihood function for the (non-Gaussian) Bernoulli noise distribution. Bernoulli noise is commonly used to model stochastic neuronal spiking, and we’ve already demonstrated that our method can handle it. We also showed in our experiments on Oja’s rule that we can handle neural trajectories with correlated signal terms, as all of the output neurons basically learn the same thing in this setting. Although we have not tried Poisson or correlated multivariate Gaussian noise models, there is no specific reason to expect that our method would fail when using their associated log-likelihood functions. The reviewer notes that the method may not “train to convergence” for other noise models, but this is a completely general concern that applies to any machine learning optimization method. Solving such a general problem is beyond the scope of our work. We agree that implementing evolutionary algorithms is a significant extension to our work that may involve unexpected challenges, but, as argued above, there are clear paths for generalization that do not require such large changes to the method.

---

> > > ### Author Response · Authors · 2023-11-20
> > > **Revised manuscript featuring direct comparisons and addressing concerns regarding generality and novelty (Part 2)**
> > >
> > > - Furthermore, we do not see the reviewer’s justification for claiming that our approach is not novel. At the time of submission, we were not aware of any work that uses neural trajectory fitting to optimize a synaptic plasticity rule. The reviewer does not point to any work that contradicts this. The most similar work seems to be Ramesh et al., which was posted online after our initial submission and has not yet been peer reviewed.

---

### Official Review · Reviewer_Xu3v · 2023-11-06

**Soundness:** 2 fair
**Presentation:** 3 good
**Contribution:** 3 good
**Rating:** 5
**Confidence:** 4

**Summary:**

The paper develops a meta-learning model for synaptic plasticity rules. They parameterize the plasticity rule as a polynomial function, or a multi-layer perceptron, of the pre-, post-activity, weight and global reward, and by gradient descent update the learning rule to minimize a loss that depends on the output trajectory. Differently from previous work, they optimize over the whole trajectory, instead of final weights/activity, and apply it also to behavioural data, instead of neural activity. One application has the loss as the distance to a target trajectory, obtained from a ground truth learning rule (Oja’s rule), which is successfully recovered. In the second application, the output activity represents the probability of taking an action on a reward-seeking task, and the target is a trajectory of actions taken. This is applied to experimental data of a fruit fly in a setting where the rewarding action changes over time. A reward-based rule of the form x*(R - E[R]) - a*w is recovered, and variations with and without E[R] or -a*w factors are discussed.

**Strengths:**

The main novel contributions of the paper are the optimization over trajectories, instead of endpoints, and fitting behavioural data.
- Optimizing over trajectories might be more efficient and appropriate when such data is available.
- Fitting a family of plasticity rules to data will strengthen our knowledge of mechanisms at work in different systems.
- It is significant to propose a new dataset that can be modelled by this approach.

Previous literature on inferring plasticity rules is properly referred to.

The simulation results seem sound.

The general methodology is clear (apart from the questions outlined below).

**Weaknesses:**

The main weaknesses are:
- Methodological choices
- Theoretical grounding of learning rules
- Limited novelty of trajectory optimization

There are methodological questions that should be clarified, listed also in the Question section below.

It is unclear why there are multiple output neurons, how they are pooled, and if they all learn the same thing. Is it the case that one needs multiple output neurons just to have enough data, and it would be equivalent to having many instantiations of the simulation with single output neurons? Do the neurons for Oja's rule all simply learn the first principal component?

Related to this point, if all neurons in the output layer learn the same thing, the study in Fig.2 with noise and sparseness might simply reflect how many "trials" are needed for a single neuron simulation. Noise in this case would just make gradients for the meta-learning less informative, needing more trials, which would be true for any learning simulation.

One main contribution of the paper relative to previous work is optimizing trajectories, instead of endpoints, but it is not clear if this is a substantial advancement. It would strengthen this point to have more results and discussion on this difference. What kind of data can this model tackle that endpoint optimization can't? Would the results be the same? Is it a matter of data efficiency? How would endpoint optimization behave in the fly dataset? A technical point: is backprop through time used?

One main contribution of the paper is the fitting of behavioral data. But there is a gap between modelling a synapse and a whole system as if the fly was a single neuron. Why can modelling a single neuron work in this case? Isn't this approach more appropriate for neural data from the relevant neurons in the fly mushroom body?

There is a lack of theoretical grounding of the learning rules studied, e.g. that the Oja rule will learn the PC1; what are the optimal learning rules for the reward-based task?; what is the effect of the weight decay term on the learning rule?

There is no discussion on the time scale inherent to the learned weight decay term. How does it relate to trial times? Related, the effect of the time scale of the E[R] is subtle and will have a different effect if the average is within a trial or over multiple trials (see Fremaux, Sprekeler, Gerstner, J Neuroscience, 2010). Related to this point, did the authors try learning the E[R] factor separately, or learning the time scale? The discussion on weight dependent vs. expected reward factor loses strength due to these limitations.

The MLP rules are included but never discussed. They should be better motivated and referred to.

Minor:

In the figures, it should show the factors in the ticks (e.g. x*R), instead of the binary parameter indexes, which are hard to decode.

**Questions:**

How are the inputs x_t sampled? N(0,1)?
How exactly are the neurons from the output layer pooled in each application?
Do all neurons in the output layer learn the same thing?

Is the loss and gradient local in time or backpropagated through time?
How does the trajectory loss compare with the end goal loss?

What do the MLP learning rules learn?

Why is the E[R] term not learned as a separate term?
How does the result depend on the time scale of E[R]?
What is the time-scale inherent to the learned weight decay term?

How should one relate the learning rule for a single neuron to the output behaviour or a whole system?

---

> ### Author Response · Authors · 2023-11-17
> **Response to reviewer Xu3v (Part 1)**
>
> Dear Reviewer,
>
> Thank you for your insightful comments and constructive feedback on our manuscript. We appreciate the opportunity to clarify and expand upon key aspects of our work. Below, we address each of your concerns in detail.
>
> **Methodological Choices and Theoretical Grounding**
>
> Our decision to include multiple output neurons in our model is primarily to demonstrate scalability. The model effectively functions with a single output neuron, but the use of multiple neurons showcases its applicability to larger, more complex networks. This is not a trivial problem as the loss is computed over the entire trajectory length and parameters are optimized with backpropagation through time. In the context of the behavioral task, the outputs are pooled through a fixed layer followed by a sigmoid function, determining the probability of accept/reject decisions. It's noteworthy that these output neurons typically follow unique trajectories. The convergence to the same outputs in the case of Oja’s rule is a specific instance, not a general trend that holds across learning rules. This aspect also underlines the versatility of our model in fitting diverse rules.
>
> We will revise the manuscript to mention the link between Oja’s rule and principal component analysis, to state that the theoretical motivation for including the E[R] term is to enable covariance-based learning that leads to operant matching, and to explain that the weight decay term could explain forgetting. However, we do not think that it is necessary to provide a theoretical rationale for the other learning rules that we consider. Instead, our focus is on fitting a broad range of arbitrary rules, without being constrained by the specific computation they implement or their optimality. This approach reflects the flexible and expansive nature of our model. Although we agree that it is interesting to relate the learning rules of single neurons to the output behavior of the whole system, that is not the topic of our paper.
>
> For the E[R] term, our experiments encompassed various time scales (5, 7, 10, 15, 20) trials, leading to consistent results across these variations. This uniformity in outcomes led us to exclude it as a learnable parameter in our optimization loop. Dynamics within the trial had no bearing on E[R].
>
> *Modelling Synaptic and System-Level Behavior:*
> Recent research, particularly by Aso et al., has demonstrated that activating a specific neuron in the fly mushroom body can influence the fly's upwind movement. This finding is crucial to our experimental setup, where odors flow and the fly's reward-driven upwind movement is a key behavior. This validates our approach of modeling single neurons, as it aligns with the observed behavior in real-world scenarios. Our approach also mimics the setting of a previous biology study by Rajagopalan et al. that we wanted to extend with our methodology. We will modify the text to explain these points and specify the exact neuron referenced in this context for clarity.
>
> Other work from Aso and collaborators has quantified how olfactory memories decay over time. In light of your comments, our revision will include an analysis of how the inferred learning rule leads weights to decay over trials, thereby leading to memory decay. We will compare the inferred learning rule to plasticity mechanisms discussed in Aso’s studies.
>
> We parameterized the learning rule with an MLP to see whether we could capture the learning rules with a general expressive model that did not match the ground-truth generative model. However, Figure 3 and Table 1 revealed that the MLP usually underperformed the Taylor series representation. We therefore decided to focus Figure 4 on the Taylor series representation, which is both more interpretable and better aligned to the known biology. We will explain these motivations and findings in the revised manuscript. For similar reasons, we did not attempt to quantify what the MLP rules learned in Figure 3. If the MLP approach had been more empirically successful, then we would have interpreted the learned rules by directly plotting the three-dimensional function of presynaptic activity, reward, and weight that they discovered. This could have easily been compared to the exact function specified by the ground-truth generative model.

---

> > ### Author Response · Authors · 2023-11-17
> > **Response to reviewer Xu3v (Part 2)**
> >
> > **Limited Novelty of Trajectory Optimization**
> > We see two possible interpretations of what the reviewer means by endpoint optimization. First, they might be imagining modeling the entire plasticity sequence as one large plasticity update, as done by Lim et al. In this case, the novelty of our approach is highly significant, because many nonlinear weight updates do not sum to be a single nonlinear update of the same mathematical form. Our model is much more aligned to the ground-truth generative process, and that allows us to recover the correct learning rule. We would be willing to add this point to the revision if the reviewer thinks it would help justify the significance of our approach.  Second, end-point optimization might simply suggest that the loss function depends only on the final states, not the full learning trajectories. We could easily do this experiment to compare trajectory optimization with this notion of endpoint optimization. However, we do not think that the outcome of this comparison has any bearing on the significance of this paper, because optimizing either loss function requires backprop through time and the framework that we introduce here to do so. No previous work has optimized a learning rule based on end points in this way. This comparison could turn out to be interesting, but adding it would not significantly improve our paper, and we would prefer to not do so.
> >
> > **Minor Issues:**
> > Our revision will add more details regarding the sampling of x_t. In brief, the inputs  x_t  in our model are designed to mimic the sparse encoding of odors by Kenyon Cells in the mushroom body. We achieve this by selecting a fixed percentage of random inputs to represent "firing" for one odor. Our experiments tested various firing values from 0.5 to 2 and consistently found stable results across these values. Additionally, we incorporate random Gaussian noise with varying variance to further simulate realistic conditions. The results we have reported include experiments with 0.1 random noise added to inputs between odors.
> >
> > We appreciate your attention to detail in the figures. We think that the compact theta notation is very useful and more accurate. Labeling the tick marks with, for example, x*R could incorrectly suggest that we are plotting that quantity. Instead, we’re plotting the coefficient that multiplies it, which we notate consistently and repeatedly with theta. Rather than changing the labels on the plots, our revision will make the notation clearer for future readers by modifying the legends to state what the significant coefficients multiply.
> >
> > We hope this response addresses your concerns and clarifies the aspects of our work. We are open to further discussions and are committed to refining our manuscript based on your invaluable feedback.

---

> > > ### Author Response · Authors · 2023-11-20
> > > **Revised manuscript, and request for additional feedback**
> > >
> > > We want to alert the reviewer that we have posted an updated manuscript that concretely addresses some key concerns that have come up in conversation with Reviewers 2 and 4. We believe that the revision we have proposed above will also address all of this reviewer’s major questions and concerns. Please let us know if there are any lingering issues that prevent the reviewer from considering updating their overall rating, as we would like to address them in the rebuttal period.

---

> > > > ### Comment · Reviewer_Xu3v · 2023-11-21
> > > >
> > > > Dear authors, thank you for the detailed answer.
> > > >
> > > > Overall, as the authors argue, the main contribution is the optimization over a trajectory, instead of final endpoints or a loss function, using backprop through time. As this is the main contribution, this method and the advantages of this method in comparison with the alternatives should be much clearer. If I'm not mistaken, the paper does not mention backpropagation through time at all. One could also optimize over trajectories without it, which might be more data efficient, with similar results.
> > > >
> > > > > In the context of the behavioural task, the outputs are pooled through a fixed layer followed by a sigmoid function, determining the probability of accept/reject decisions. It's noteworthy that these output neurons typically follow unique trajectories. The convergence to the same outputs in the case of Oja’s rule is a specific instance, not a general trend that holds across learning rules. This aspect also underlines the versatility of our model in fitting diverse rules.
> > > >
> > > > How are output neurons pooled exactly? Adding more output neurons does not increase scalability if they are all doing the same thing, it only increases data availability, which might be necessary because the method is more noisy and data-hungry. Why do output neurons learn unique trajectories? Shouldn't they all learn the same, if they are pooled to give the same output? Showing that your method works for a more specific case, does not show the versatility to more general cases.
> > > >
> > > > > However, we do not think that it is necessary to provide a theoretical rationale for the other learning rules that we consider. Instead, our focus is on fitting a broad range of arbitrary rules, without being constrained by the specific computation they implement or their optimality. This approach reflects the flexible and expansive nature of our model.
> > > >
> > > > To argue that the method is more powerful than others, and can infer plasticity rules without priori knowledge, then it should be shown that the method works without such knowledge, e.g. with more polynomial terms than expected. On the contrary, the paper seems to use prior knowledge at multiple points to constrain the number of factors, as in the specific inclusion of E[R] and w, the fixed time constant of E[R], and the exclusion of y.

---

> ### Author Response · Authors · 2023-11-21
> **Clarifications and updates**
>
> We thank the Reviwer for the follow-up questions and concerns. We address each of them below. With these clarifications and planned changes, we believe that we will have been able to address all of the reviewer’s major concerns. Please let us know if there are any lingering issues that prevent the reviewer from considering updating their overall rating.
>
> 1.
>
> Our main contribution is indeed optimization over a trajectory with backpropagation through time (BPTT). To be precise, however, we still optimize a loss function, although it is defined as the error between the data and model trajectories, rather than a functional relationship as in previous work [Confavreaux et al. 2021, Tyulmankov et al. 2022]. To optimize this loss function we use BPTT, which was implicit in the definition of the loss function (Eq. 3) and its derivative (Eq. 4) due to the time dependency. We agree that this should be made more clear and explicit, and we will update the text accordingly.
>
> It might be helpful to distinguish between the optimization *problem* and the used optimization *technique* to solve it. Our approach is fundamentally different from alternative setups in terms of the optimization problem. We provide a discussion and comparison in the "Related Work" section. The only other work we are aware of that uses neural trajectories to fit a plasticity rule is [Ramesh et al. 2023], which was only published to *bioRxiv* after our initial submission. We will include a qualitative comparison in the final version of our manuscript nevertheless. In terms of comparison to other optimization techniques, the choice of optimizer is an implementation detail that we could revisit in future work. Indeed, alternatives for computing gradients such as real-time recurrent learning (RTRL) can be used, or approximations to the gradient such as truncated BPTT, or even non-gradient based methods such as evolutionary algorithms.
>
> However, any approximations (whether approximations to the gradient or using only endpoints in the loss) will be strictly less data-efficient (although may be more computationally- and memory-efficient), since we would be ignoring long-term dependencies between the parameters and data. As such, we opted for the most powerful methodology because we wanted it to work for arbitrary rules, some of which might be highly nonlinear and benefit from long time dependencies more than others (see below).
>
>
> 2.
>
> The reviewer asked several related questions. Our responses are:
> * Output neurons are uniformly summed (weights equal to 1), and a logistic sigmoid function is applied to that sum to generate a probability of choosing the current odor.
> * Our discussion of  “scalability” refers to our implementation’s ability to handle large networks, rather than any comment about data efficiency.
> * In the general case, learning rules define nonlinear dynamical systems that can depend sensitively on their initial conditions as well as the particular noise instantiation. Because of this, the output neurons do not necessarily follow identical trajectories – the pooled behavioral readout is then a linear combination of their individual (unique) outputs. As such, additional output neurons do offer additional information about the plasticity rule beyond simple data availability.  However, as with Oja’s rule (which learns weights that align with the first principal component), it may be the case that output neurons behave similarly. Nevertheless, Table 1 shows that the method works for many different learning rules beyond Oja’s rule, and it is inaccurate to say that we’re generalizing because our method “works for a specific case.”

---

> ### Author Response · Authors · 2023-11-21
> **Clarifications and updates (part 2)**
>
> 3.
>
> The problem of hypothesis generation is indeed an important and difficult one. Here, we simply choose a hypothesis using prior knowledge established in other work. We demonstrate that our method is reliable for a given hypothesis class. Generating a hypothesis (i.e. the functional family as well as choosing the appropriate inputs) can be done through what has been referred to as "Box's loop" [Blei, 2014]. We iterate on hypotheses and do model validation for each one until we reach satisfactory convergence (or a deadline).
>
> We do not claim that plasticity rules can be inferred without any prior knowledge. The most important prior knowledge that underlies this work is that plasticity rules in biology can only local information, such as presynaptic and postsynaptic activity, current synaptic weight, and global dopaminergic reward signals. This, by definition, constrains the plasticity rules that we fit to be biologically plausible. We note that this prior knowledge is far less restrictive than prior knowledge of the circuit’s *computation,* which was assumed in the previous work of Confavreaux et al. and Tyulmankov et al. Given sufficient data and computational power, one could consider a sufficiently broad hypothesis which includes all the possible inputs and allow the optimization routine to decide which ones should be used and in what way. For instance, consider
> $$
> g_\theta = \sum_{\alpha,\beta,\Phi,\gamma} \theta_{\alpha,\beta,\Phi,\gamma} x_j^\alpha r^\beta y_i^\Phi w_{ij}^\gamma
> $$
> We did not explore this learning rule in our submission because it involves significantly more coefficients that must be fit, and hypothetical dependencies of the mushroom body’s learning rule on $y$ can be excluded on biological grounds. However, to address the reviewer’s concerns, we will try fitting this more complex learning rule in our revision to see whether the coefficients can still be recovered.

---

### Meta-Review · Area_Chair_8pM6 · 2023-12-05

**Metareview:**

The paper studies meta-learning  for synaptic plasticity rules. They parameterize the plasticity rule as a polynomial or a multi-layer perceptron, and use gradient descent to update the learning rule to minimize a loss that depends on the output trajectory of the system. They differ from previous work int hat they a) optimise her the whole trajectory (rather than final states), and also in that they apply the approach to behavioural data (not just neural data).

The paper addresses an interesting problem, and reviewers generally agreed that it was sound and that there was no fundamental flaw in it.

However, concerns remained about the applicability, generality and scalability of the new method, as well as about the interpretability of the results. In particular, it seems questionable to  directly fit a model of synaptic dynamics to  whole animal bahvior, and this leap between description levels was not well motivated or justified.

**Justification For Why Not Higher Score:**

see above

**Justification For Why Not Lower Score:**

see above

---

### Decision · Program_Chairs · 2024-01-16

Reject